# Stable Aqueous Colloidal Solutions of Nd^3+^: LaF_3_ Nanoparticles, Promising for Luminescent Bioimaging in the Near-Infrared Spectral Range

**DOI:** 10.3390/nano11112847

**Published:** 2021-10-26

**Authors:** Alexandr Popov, Elena Orlovskaya, Artem Shaidulin, Ekaterina Vagapova, Elena Timofeeva, Leonid Dolgov, Lyudmila Iskhakova, Oleg Uvarov, Grigoriy Novikov, Mihkel Rähn, Aile Tamm, Alexander Vanetsev, Stanislav Fedorenko, Svetlana Eliseeva, Stephane Petoud, Yurii Orlovskii

**Affiliations:** 1Prokhorov General Physics Institute of the Russian Academy of Sciences, 119991 Moscow, Russia; avpopov@lst.gpi.ru (A.P.); lenao@Lst.gpi.ru (E.O.); shatarte@yandex.ru (A.S.); ekaterina.vagapova@ut.ee (E.V.); elena16051997@mail.ru (E.T.); ldisk@fo.gpi.ru (L.I.); uvarov@kapella.gpi.ru (O.U.); alexander.vanetsev@ut.ee (A.V.); 2Institute of Physics, University of Tartu, W.Ostwaldi Str. 1, 50411 Tartu, Estonia; leonid.dolgov@ut.ee (L.D.); mihkel.rahn@ut.ee (M.R.); aile.tamm@ut.ee (A.T.); 3Institute of Cyber Intelligence Systems, National Research Nuclear University MEPhI, 115409 Moscow, Russia; ggnovikov@mephi.ru; 4Voevodsky Institute of Chemical Kinetics and Combustion SB RAS, Institutskaya Str. 3, 630090 Novosibirsk, Russia; fedorenk@kinetics.nsc.ru; 5Center for Molecular Biophysics CNRS, UPR 4301, Rue Charles-Sadron, CEDEX 2, 45071 Orléans, France; svetlana.eliseeva@cnrs-orleans.fr (S.E.); stephane.petoud@inserm.fr (S.P.)

**Keywords:** Nd^3+^: LaF_3_, nanoparticles, rare earth aqueous colloidal solutions, hydrothermal microwave synthesis, NIR fluorescence, radiative lifetime, Nd–Nd fluorescence self-quenching, Nd–OH Förster quenching

## Abstract

Two series of stable aqueous colloidal solutions of Nd^3+^: LaF_3_ single-phase well-crystallized nanoparticles (NPs), possessing a fluorcerite structure with different activator concentrations in each series, were synthesized. A hydrothermal method involving microwave-assisted heating (HTMW) in two Berghof speedwave devices equipped with one magnetron (type I) or two magnetrons (type II) was used. The average sizes of NPs are 15.4 ± 6 nm (type I) and 21 ± 7 nm (type II). Both types of NPs have a size distribution that is well described by a double Gaussian function. The fluorescence kinetics of the ^4^F_3/2_ level of the Nd^3+^ ion for NPs of both types, in contrast to a similar bulk crystal, demonstrates a luminescence quenching associated not only with Nd–Nd self-quenching, but also with an additional Nd–OH quenching. A method has been developed for determining the spontaneous radiative lifetime of the excited state of a dopant ion, with the significant contribution of the luminescence quenching caused by the presence of the impurity OH– acceptors located in the bulk of NPs. The relative quantum yield of fluorescence and the fluorescence brightness of an aqueous colloidal solution of type II NPs with an optimal concentration of Nd^3+^ are only 2.5 times lower than those of analogous Nd^3+^: LaF_3_ single crystals.

## 1. Introduction

Currently, an important scientific goal is to design aqueous colloidal solutions (ACS) of nanoscale fluorophores based on dielectric crystals doped with rare earth (RE) ions. These luminescent nanoparticles (NPs) combine a number of unique properties, including narrow spectral absorption and emission lines and long lifetimes of their excited states, allowing for time detuning from autofluorescence of biological tissues and high photo- and physicochemical stability [1,2,3,4,5]. All these advantages make it possible to use them in biology and medicine as in vitro and in vivo luminescent probes for visualization in the visible (VIS) and near infrared (NIR) spectral ranges in the first (0.75–0.95 µm) and second (1.0–1.2 µm) optical “windows” of transparency of biological tissues (“biological windows”) [6,7,8,9]. Bioimaging with NIR radiation is advantageous compared to visible light because it can penetrate biological tissue to a great depth (one centimeter) [10,11,12,13,14,15] without causing photoinduced cytotoxicity. In addition, NIR signals can be discriminated from the natural fluorescence of biological tissues (autofluorescence) due to the increased signal to noise ratio, which provides more specific and sensitive detection.

Due to the closeness of the radii of the lanthanum and neodymium ions, the crystal matrix of LaF_3_ is convenient for doping, since it allows up to 100% replacement of lanthanum ions with neodymium ions [16,17]. In addition, the ^4^F_3/2_ → ^4^I_9/2_, ^4^I_11/2_ transitions of the neodymium ion ensure luminescence in the first and second biological windows [18].

An additional advantage of the LaF_3_ crystal matrix doped with Nd^3+^ ions is the large value of the ratio of the Judd–Ofelt intensity parameters Ω_4_/Ω_6_ [19]. It provides a relatively high luminescence branching ratio *β* at the ^4^F_3/2_ → ^4^I_9/2_ transition of the Nd^3+^ ion, i.e., in the first biological window, almost equal to the luminescence branching ratio on the well-known ^4^F_3/2_ → ^4^I_11/2_ laser transition in the second biological window. The luminescence in the first biological window can be monitored with less expensive detectors than in the second. The relatively low value of the parameter Ω_6_ in the LaF_3_ crystal [19] also provides less effective self-quenching of the ^4^F_3/2_ level of Nd^3+^ ions and weakened quenching of the luminescence caused by vibrations of the OH– acceptors remaining in the form of defects in the crystal lattice of LaF_3_, as the result of their synthesis in an aqueous medium. Consequently, Nd^3+^: LaF_3_ exhibits weaker luminescence self-quenching and quenching than in many other crystals and NPs doped with Nd^3+^ ions [19]. Although until recently there was no understanding of the physical reasons explaining the advantages of this crystal over other similar crystals, Nd^3+^: LaF_3_ NPs and their aqueous solutions have nevertheless been developed for more than 15 years for optical bioimaging in the first [20,21] and second biological windows [16]. Such NPs can simultaneously combine optical and magnetic nanoheaters [22,23,24], as well as spectral or kinetic nanothermometers [24,25,26]. In addition to ensuring successful medical use, these NPs should have optimal sizes, the ability to form stable aqueous colloidal solutions, chemical and biological inertness, and low toxicity [27].

To analyze the degree of luminescence quenching by unexcited impurity ions and uncontrolled acceptors, it is acceptable to use such a value as the relative quantum yield of luminescence $\varphi /\varphi_{0}$ [28]. If the law of luminescence decay I(t) under delta-pulse excitation is known, then the relative quantum yield of luminescence is determined by the following expression:(1)$$\frac{\varphi }{\varphi_{0}}=\frac{1}{\tau_{D}}\int_{0}^{\infty }I(t)dt=\frac{1}{\tau_{D}}\int_{0}^{\infty }N(t)\exp\left(-\frac{t}{\tau_{D}}\right)dt$$

Here, $\varphi_{0}$ and $\tau_{D}$ are the quantum yield and luminescence lifetime of the donor in the absence of quenching due to inter-center energy transfer, and N(t) and φ are kinetics of impurity quenching of luminescence and the quantum yield in the presence of inter-center energy transfer.

Usually, in the absence of energy transfer, the number of excited optical centers decreases exponentially with a characteristic decay rate equal to the sum of the radiative and nonradiative intra-center decay rates $1/\tau_{D}=1/\tau_{R}+1/\tau_{N}$. Therefore, $\varphi_{0}$ is equal to the product of the radiative decay rate $1/\tau_{R}$ by the luminescence lifetime $\tau_{D}$ in the absence of energy transfer.
(2)$$\varphi_{0}=\frac{1}{\tau_{R}}\int_{0}^{\infty }\exp\left(-\frac{t}{\tau_{D}}\right)dt=\frac{\tau_{D}}{\tau_{R}}=\frac{\tau_{N}}{\tau_{N}+\tau_{R}}$$

Similarly, the quantum yield of impurity quenching of luminescence φ is
(3)$$\varphi =\frac{1}{\tau_{R}}\int_{0}^{\infty }N(t)\exp\left(-\frac{t}{\tau_{D}}\right)dt$$

Combining (2) and (3), we obtain Formula (1). For metastable levels $\tau_{D}\approx \tau_{R}$, and therefore $\varphi_{0}\approx 1$. As a result, we obtain that the Formulas (1) and (3) can be written in the form
(4)$$\frac{\varphi }{\varphi_{0}}\approx \varphi \approx \frac{1}{\tau_{R}}\int_{0}^{\infty }N(t)\exp\left(-\frac{t}{\tau_{R}}\right)dt$$

The fluorescence brightness is determined by the product of the absolute concentration $n_{D}$ of Nd^3+^ ions and the relative quantum yield of impurity quenching of luminescence equals φ for metastable levels:(5)$$\nu =n_{D}\varphi$$

This value is actually proportional to the radiation intensity of the donor at a concentration $n_{D}$ and is widely used in the literature [29,30].

In the literature, we have not come across an accurate experimental determination of the relative quantum yield of the luminescence of ACS of Nd^3+^: LaF_3_ NPs synthesized by various methods. A fairly high relative quantum yield of 48% [31] was determined in well-crystallized 3 at.% Nd^3+^: LaF_3_ NPs about 30 nm in size, synthesized by the hydrothermal method from an aqueous ethanol solution and redispersed in an anhydrous solvent. The quantum yield was determined from the ratio of the decay time of luminescence by a factor of *e* $\left(\tau_{e}=369\;\mu s\right)$ to the radiation lifetime of the ^4^F_3/2_ level $\left(\tau_{R}=753\;\mu s\right)$, which was obtained from the absorption spectra of this colloid using the Judd–Ofelt theory. However, it is doubtful whether the refractive index of a colloidal solution with different refractive indices of NPs and a solvent was correctly taken into account when calculating the radiative lifetime. It seems to us that it is underestimated, since the refractive index of the proposed complex solvent is lower than that of Nd^3+^: LaF_3_ NPs. Below, we will dwell on this issue in more detail. Similarly, [32] mentions the achievement of the relative quantum yield of NIR luminescence $\varphi /\varphi_{0}=95\%$, measured in nanocomposites obtained by dispersing 0.5 at% Nd^3+^: LaF_3_ NPs about 10 nm in size, synthesized by the solvothermal method, with the encapsulation of these NPs in a polymer shell. Such a high value of the quantum efficiency of luminescence in the NIR spectral range, calculated from the ratio $\tau_{e}=800\;\mu s$ at 0.5 at% Nd^3+^ to the $\tau_{e}=846\;\mu s$ of the ^4^F_3/2_ level of the Nd^3+^ ion (estimated by the Judd–Ofelt theory using absorption spectra) indicates the almost complete absence of the luminescence quenching of the Nd^3+^ ions by OH− acceptors in the bulk of NPs obtained by a non-aqueous method. However, again, when calculating the radiation lifetime, the refractive index of a nanocomposite consisting of the NPs surrounded by a polymer, which have different refractive indices, was not correctly taken into account. The value is also underestimated, since the refractive index of the polymer is lower than that of Nd^3+^: LaF_3_ NPs. In [25], which used the thermal lens method [33] in LaF_3_: Nd^3+^ NPs, it was found that at a low concentration of Nd^3+^ ions, the relative quantum yield of luminescence reaches 80%, and at a concentration of 20 at% it decreases to 20% [25]. In [34], the limitation of the inapplicability of the thermal lens method for estimating the relative quantum yield of luminescence of aqueous colloidal solutions of NPs is discussed in detail. This method gives extremely overestimated values of the quantum yield by almost an order of magnitude.

A method for determining the relative fluorescence quantum yield $\varphi /\varphi_{0}$ (Equation (4)) of the ^4^F_3/2_ excited state of the Nd^3+^ ion, based on the ratio of the area under the measured luminescence decay curve *I_meas_(t)* and the lifetime $\tau_{R}$ of the radiative spontaneous decay of this state, determined at the late stage of the luminescence kinetics of the powder and ACS of 0.1 at% Nd^3+^: LaF_3_ NPs, was applied in [34] and [19], respectively. Note, that the intra-center quantum yield of luminescence for the ^4^F_3/2_ metastable state tends toward unity ($\varphi_{0}\to 1$) in response to an insignificant intra-center nonradiative relaxation due to the large energy gap (ΔE ≈ 5000 cm^−1^) [35] between the ^4^F_3/2_ level and the next lower energy level ^4^I_15/2_. As stated above, the fluorescence quenching of dopant ions in Nd^3+^: LaF_3_ NPs depends on two independent donor–acceptor processes of transfer of electronic excitation energy from the ^4^F_3/2_ excited state of the Nd^3+^ ion. Namely, concentration self-quenching due to the cross-relaxation through the ^4^I_15/2_ electronic state [16] and on quenching, caused by anharmonic vibrations of molecular groups. It is known that hydroxyl OH−groups are one of the most effective quenchers of luminescence in the near-IR range [36].

At a very low concentration of Nd^3+^ (0.1 at% Nd^3+^), the efficiency of the self-quenching process of the luminescence of Nd^3+^ ions in crystals is close to zero. Therefore, the maximum relative quantum yield of luminescence from the ^4^F_3/2_ level of the Nd^3+^ ion, close to unity, is observed in a LaF_3_ single crystal doped with a low concentration of Nd^3+^ ions [17]. On the contrary, the contribution of luminescence quenching on OH–acceptors in NPs can be significant, even at a low concentration of Nd^3+^ ions, which leads to a much lower quantum yield in NPs. The value of φ determined in the dried powder of 0.1 at.% Nd^3+^: LaF_3_ NPs synthesized by the hydrothermal method with microwave treatment with Proxanol-268 is about 30% [34]. Consequently, the contribution of the Nd–OH donor–acceptor quenching to the overall relaxation rate is about 70%, which indicates its high efficiency in Nd^3+^: LaF_3_ NPs synthesized from aqueous solutions. Thus, the problem of NIR luminescence quenching of Nd^3+^ ions NPs is fundamental, since most of the synthesis methods, which can result in their stable aqueous colloidal solutions, are carried out either in water or in other media containing OH−groups. At the same time, methods for the synthesis of NPs doped with rare earth ions largely determine their physicochemical and luminescent properties. Currently, the most simple and widespread methods of synthesis are co-precipitation [34,37,38], solvothermal [39,40,41], and hydrothermal [34,42,43]. Each of these methods, used to synthesize stable colloids of Nd^3+^: LaF_3_ NPs with the aim of creating efficient nanosized phosphors on their basis, have their own advantages and disadvantages.

The co-precipitation method is the simplest and fastest way for obtaining NPs, since it does not require extreme synthesis conditions and/or expensive equipment and is carried out using organic or inorganic solvents. During the synthesis of Ln^3+^: LaF_3_ NPs (Ln^3+^ = Eu^3+^, Er^3+^, Nd^3+^, Ho^3+^) in an ethanol–aqueous medium at 60 °C [37] using a surface-modifying agent (ammonium di-n-octadecyl dithiophosphate), highly crystalline luminescent Ln^3+^: LaF_3_ NPs with a size of 7–10 nm were obtained. However, in order to obtain a stable aqueous colloid, the hydrophobic surface of these NPs requires an additional complex modification procedure to transform them into a hydrophilic entity. High-temperature annealing (up to 500 °C for 90 min) of Nd^3+^: LaF_3_ NPs synthesized by a conventional co-precipitation method in an aqueous medium [38] improved the luminescence properties and crystal structure of these NPs. However, after annealing, they significantly aggregated with an increase in average size from 12 nm to 40 nm, which is unlikely to facilitate their redispersion in water.

In the solvothermal method of synthesis from organic solvents, the initially co-precipitated gel is treated at high temperature and pressure to obtain non-aggregated NPs with a narrow size distribution. These NPs contain much fewer defects in the form of OH-groups and water molecules. Therefore, the quenching of the luminescence of these NPs is much weaker than that obtained by water synthesis methods. On the other hand, such NPs almost always have a hydrophobic surface, which necessitates the modification of their surface using surfactants, leading to an increase in their toxicity and complication of the technology for the production of fluorescent probes based on them [39,40].

As a result of the hydrothermal synthesis, as a variation of the solvothermal synthesis using water as a solvent, non-toxic, well-crystallized colloidal Nd^3+^: LaF_3_ NPs with a hydrophilic surface are synthesized, capable of forming stable aqueous colloidal solutions. However, at the same time, these NPs have a wider size distribution and contain an increased concentration of OH−groups in the crystalline matrix of LaF_3_ and residual water in the mesopores, which leads to an increase in the quenching of the luminescence of Nd^3+^ ions, as compared to NPs of solvothermal synthesis. The formation of NPs of low-soluble compounds under hydrothermal conditions is usually described by the dissolution–crystallization mechanism [44,45]. When a freshly precipitated gel of a poorly soluble compounds is exposed to hydrothermal conditions, it undergoes a collective recrystallization process, which largely determines the size distribution of nanoparticles, the stability of the final colloid, and the degree of crystallinity and the defectiveness of the NPs. [44,45]. Thus, the hydrothermal method makes it possible to obtain hydrophilic, partially agglomerated, highly crystalline, and compositionally homogeneous NPs of low-water-soluble compounds. An increase in the morphological homogeneity of NPs, as well as a reduction in the duration of hydrothermal treatment to obtain well-crystallized materials, is possible with the use of microwave heating [46,47]. At the same time, hydrothermal microwave treatment (HTMW) can be used both for direct synthesis from the solution and for the crystallization of pre-precipitated gels [48].

As shown in [34], hydrophilic Nd^3+^: LaF_3_ NPs obtained by co-precipitation do not meet the necessary requirements due to strong self-quenching of luminescence in the Nd*–Nd pairs and the Nd*–OH quenching when acceptors are in the bulk of NPs. Therefore, this method is supplemented by hydrothermal microwave treatment in an aqueous medium using the surfactant Proxanol-268. It was shown that the use of HTMW can significantly enhance the luminescence and improve the crystalline properties of Nd^3+^: LaF_3_ NPs. Analysis of the luminescence properties of these NPs showed that they have a much lower degree of defectiveness and a much higher fluorescence brightness in the near IR spectral range, due to weaker luminescence quenching. Thus, the co-precipitation method, supplemented by the HTMW treatment, is one of the most convenient and promising methods for obtaining highly crystalline luminescent NPs for use in medicine [48].

In this work, we continued the study (begun in [19]) of the concentration dependence of the relative quantum yield and the fluorescence brightness of the impurity luminescence of Nd^3+^ ions in long-term stable aqueous colloidal solutions of Nd^3+^: LaF_3_ NPs synthesized without surfactants. Hydrothermal synthesis of colloids was carried out on two different devices capable of microwave heating, differing in the number of magnetrons.

The aim of this work is to maximize the luminescence brightness, in the NIR spectral range, of stable aqueous colloidal solutions of Nd^3+^: LaF_3_ NPs synthesized by the hydrothermal microwave method by reducing fluorescence quenching, which will make them applicable for luminescent imaging in the first transparency window of biological tissues.

## 2. Materials and Methods

### 2.1. Synthesis

Two series of time-stable aqueous colloidal solutions of NPs *x* at% Nd^3+^: LaF_3_, where *x* = 0.1, 1, 2, 3, 4, 6, 12 for the first series (NPs type I) and 0.1, 1, 2, 3, 4, 6, 8, 10, 12 (NPs type II) for the second, were synthesized by hydrothermal-microwave treatment (HTMW) of freshly precipitated gels at 200 °C for 2 h in different HTMW devices: speedwave Four (Berghof Products+ Instruments GmbH, Eningen unter Achalm, Germany) with one magnetron (2.45 GHz, 1 kW maximum output power) (samples of type I) and speedwave XPERT (Berghof Products+ Instruments GmbH, Eningen unter Achalm, Germany) with two magnetrons (2.45 GHz, 2 kW maximum output power) (samples of type II).

The initial reagents used in the synthesis without any further purification include Nd(NO_3_)_3_·5H_2_O (Aldrich, 99.999% purity), La(NO_3_)_3_·6 H_2_O (99.999%), NH_4_F (>98%), and KF > 99% purity (Sigma-Aldrich, Stockholm, Sweden AB). For the synthesis of aqueous colloidal solutions of Nd^3+^: LaF_3_ NPs doped with (0.1–12 mol.%) Nd^3+^ ions, La(NO_3_)_3_·6H_2_O (0.4995–0.44 mmol) and Nd(NO_3_)_3_·5H_2_O (0.0005–0.06 mmol) were dissolved in deionized water (15 mL). The solution of rare earth salts was added dropwise to the NH_4_F solution (5 mmol) in deionized water (25 mL) under vigorous stirring. The freshly precipitated gels were diluted with deionized water (10 mL) and left stirring for 15 min. The resulting solutions were transferred into a 100 mL Teflon autoclave and placed under microwave irradiation for 2 h at 200 °C using a speedwave Four or speedwave XPERT laboratory device. After they were cooled, they were centrifuged using a Thermo Scientific Heraeus Multifuge X1 or Hermle Z326 device correspondingly and washed several times with deionized water. The resulting precipitates were redispersed in deionized water using ultrasonication.

### 2.2. Nonoptical Characterization

X-ray diffractograms (XRD) of both powders were detected using Bruker D2 Phaser powder X-ray diffractometer with CuKα radiation. Processing of the results, the phase analysis of the powders, and lattice parameters refinement were performed using software package DIFFRACplus (TOPAS 4.2.0.2).

Samples of type I were studied at the Institute of Physics of the University of Tartu (Estonia) using transmission electron microscopy (TEM) analysis, and those measurements were performed in the scanning mode (STEM) at 200 kV using a Cs-probe-corrected transmission electron microscope (FEI Titan Themis 200, ThermoFisher Scientific, Hillsboro, OR, USA). Powders in solutions were diluted in ethanol and ultrasonicated. The colloid was placed on a TEM copper grid with carbon film and dried for several hours. Energy dispersive X-ray spectroscopy (EDX) signal of the NPs was collected with SuperX silicon drift detectors (Bruker, Billerica, MA, USA) to measure element concentrations. Quantitative analysis was performed with Cliff–Lorimer method for F with K-line and for La and Nd with L-line using Bruker Esprit software. It provides accurate enough elemental analysis with high spatial resolution.

Samples of type II were studied at the GPI RAS (Moscow). The transmission electron microscopy (TEM) and STEM images of the samples of type II were taken with Zeiss Libra 200 FT HR microscope under accelerating voltage 200 kV. The colloid was placed on a TEM copper grid with carbon film and dried for several hours. EDX spectrometer was controlled by EDS Aztec OXFORD software. Processing of TEM images for calculation of size distribution was carried out using the ImageJ program. The statistics for each sample consisted of about 1000 NPs.

### 2.3. Optical Characterization

The study of the spectral and kinetic characteristics of NIR luminescence of aqueous colloidal solutions of Nd^3+^: LaF_3_ NPs, depending on the concentration of Nd^3+^ ions, was carried out on similar experimental setup at the Institute of Physics of the University of Tartu and GPI RAS in Moscow. In Tartu, the samples of type I were excited in the spectral range of 564–590 nm into ^4^G_5/2_ level of Nd^3+^ ion by tunable pulsed Rhodamine 6G dye laser DL-Compact (Estla Ltd., Tartu, Estonia) with laser line width Δ*λ* = 0.0065 nm at full-width-half-maximum (FWHM), pumped by the second harmonics of Nd:YAG (model LQ215, f = 20 Hz, pulse duration 5 ns, Solar laser systems, Minsk, Belarus), or Continuum Sunlite OPO system PL 9010, TRP with EX OPO frequency extension module (signal 405–705, idler 715–1750 nm, laser line width Δ*λ* = 0.003 nm at FWHM) pumped by second harmonics of Continuum YAG: Nd^3+^ laser with seeder (f = 20 Hz, pulse duration 7 ns). The wavelength of excitation was controlled by the wavelength meter WS 5 (HighFinesse, Graefelfing/Munich, Germany/Ångstrom Ltd., Novosibirsk, Russia) with an accuracy of 0.001 nm. The near infrared luminescence of the sample was focused by the condenser on the entrance slit of the Shamrock 303i spectrometer (Andor, Oxford Instruments, Abingdon-on-Thames, UK) with 1200 grooves per mm grating with linear inverse dispersion of 2.4 nm/mm. The BLP01-808R-25 edge-filter (Shamrock) was placed at the front slit of the monochromator to limit the entrance of stray light caused by laser radiation. The fluorescence was detected with the gated Andor Technology iCCD camera iStar DH320T-18H-13 with a pixel size of 26 µm and with Peltier cooling system. In Moscow, the fluorescence kinetics of the samples of type II was measured with excitation by a pulsed Al_2_O_3_-Ti laser LOTIS-TII LS-2134-LT40 (Lotis, Minsk, Belarus) (*f* = 10 Hz, *t*_pulse_ = 8–30 ns) into the ^4^F_5/2_ level of the Nd^3+^ ions. The fluorescence that originated from the ^4^F_3/2_ level of Nd^3+^ ions was dispersed by MDR 23 monochromator (LOMO, St. Petersburg, Russia) with 0.1 nm spectral resolution. The longpass filter FEL0850 (Thorlabs, Newton, NJ, USA) was attached to its front slit to block the laser radiation. The fluorescence kinetics of the samples of type I was detected by a Hamamatsu PMT 6240-02 (Hamamatsu Photonics, Naka Ward, Sunayamacho, Japan) and the samples of type II by a Hamamatsu PMT R13456P in gated photon counting mode with a multi-channel analyzer Fast Comtec P7882 (FAST ComTec Communication Technology GmbH, Oberhaching, Germany) with time resolution of 100 ns (samples of type I) and multichannel scaler (MCS) Timeharp 260 (PicoQuant GmbH, Berlin, Germany) with subnanosecond time resolution (samples of type II). Constant fraction discriminators of the NIM standard (ORTEC/AMETEK, Oak Ridge, TN, USA) were used for accurate timing of triggering and counting pulses of fluorescence signal of samples of type II. The fluorescence of both types of samples was detected at the ^4^F_3/2_ → ^4^I_9/2_ transition of Nd^3+^ ions. To obtain the fluorescence kinetics undistorted by various nonlinear processes, usually defined as up-conversion [49] in the case of excitation into the ^4^F_5/2_ level in the study of samples of type II, we decreased the energy of the laser excitation pulse to a value when further decrease in energy affected only the fluorescence intensity, but not the kinetics itself. Since the NIR luminescence of type I NPs was excited through the high-lying ^4^G_5/2_ level, the effect of up-conversion on the excitation of the ^4^F_3/2_ metastable level was insignificant due to longer multiphonon relaxation times at higher levels compared to the exciting laser pulse.

The excitation spectrum in the range of 300–925 nm was recorded in the Center for Molecular Biophysics CNRS Orléans, France. A sample was placed into a quartz capillary with 2 mm interior diameter (i.d.) using a custom-designed Horiba Scientific Fluorolog 3–22 spectrofluorometer equipped with integrated sphere and visible photomultiplier tube (PMT) (220–950 nm, R13456; (Hamamatsu Photonics, Naka Ward, Sunayamacho, Japan) and a NIR PMT (950–1650 nm, H10330–75; Hamamatsu) upon excitation with a continuous Xenon lamp. The excitation spectrum was corrected for the instrumental functions.

## 3. Results and Discussion

### 3.1. Structure of the Nanoaprticles

According to the XRD (Appendix A), all synthesized samples are pure LaF_3_ phase with a fluorcerite structure (space group P3c1, ICDD PDF 78-1864) with a centrosymmetric unit cell.

Analysis of the projections of NPs from TEM images showed that after drying on a carbon film, drops of colloidal solution of Nd^3+^: LaF_3_ NPs can partially agglomerate and partially remain isolated (Figure 1a,b). In general, the NPs are well crystallized and partially faceted (Figure 1a–d). In high resolution HR STEM images (Figure 1c,d), the projections of NPs show bright areas of mesopores, which are located in the volume or on the surface of NPs and are probably filled with a mother liquor.

Elemental analysis (Appendix A) has shown that, in both types of NPs, the ratio of the main chemical elements (La/F) remains constant and very close to the stoichiometric ratio. Oxygen and dopant element (Nd) were also detected. However, their correct quantification in both types of NPs was impossible due to low content.

To carry out a statistical analysis of the size distribution of NPs, we assumed that the most suitable geometric figure for approximating the projection shape of most NPs is an ellipse (Figure 2). We obtained size distributions of both types of NPs (Figure 3) in accordance with the parameters of large (*D*) and small (*d*) diameters of an ellipse (Figure 2), fitted around the projection of the nanoparticle.

All arrays of distributions of NPs of both samples (Figure 3) are poorly described by one normal Gaussian distribution function, but at the same time, they are well approximated by a sum of two normal distribution functions (parameter R^2^ > 0.99) (Figure 3a,b). In this regard, we assume that both types of NPs contain two fractions: a finely dispersed fraction (Fine, index F), which corresponds to NPs with smaller sizes, and the second coarse fraction (Coarse, index C), corresponding to larger NPs.

The parameters *D*_F_, *D*_C_ and *d*_F_, *d*_C_, respectively, in each distribution (Figure 3, Appendix A) are the positions of the maxima of two Gaussian functions, the sum of which describes the size distribution of NPs in accordance with the large (*D*) and small (*d*) diameters of the approximating ellipse. The volume ratio parameter is the ratio of the specific contributions of these two Gaussian functions, determined by the ratio of the areas under them in the same distribution (Figure 3, Appendix A). The fraction with smaller projection sizes of NPs, characterized by the parameters *D*_F_ and *d*_F_, has a narrower size distribution than those with the larger sizes (Appendix A). This fraction of NPs appears to be formed as a result of the primary crystallization of the gel from solution. The fraction of larger NPs, which is described by the *D*_C_ and *d*_C_ parameters, is apparently the result of recrystallization and aggregation of primary NPs during HTMW treatment. Thus, in spite of the same general conditions of synthesis, the geometry of the speedwave XPERT setup leads to an increase in the growth and recrystallization of NPs of the sample of type II (Figure 3, Appendix A), which leads to an increase in their crystallinity and, consequently, to a decrease in quenching of NIR luminescence of these NPs. The reason for the intensification of growth during HTMW treatment in a setup with two magnetrons requires additional research. Apparently, this is due to the intensity and distribution of the electromagnetic field during treatment of the autoclaved sample. It should be noted that, in different types of samples of aqueous colloidal solutions of Nd^3+^: LaF_3_ NPs, the ratio of these two fractions changes only to a small extent. In the sample of type I, the contribution of the fraction of small NPs is slightly more than 50% (53–57%) (Figure 3a, Appendix A), while in the sample of type II this contribution varies from 49 to 64% (Figure 3b, Appendix A). In this case, the fraction of small NPs of the sample of type I is finer than that finely dispersed fraction of NPs of the sample of type II. The same applies to the coarse fraction of NPs of both types.

The <*D*_sphere_> values are calculated considering the average specific contribution of each fraction of NPs, which is characterized by the average diameter <*D*_N sphere_> (*n* = F, C) (Appendix A). In turn, the average diameter <*D*_N sphere_> is calculated from the condition of equality of the volumes of a sphere, with such a diameter and model ellipsoids obtained by rotating an ellipse with diameters *D*_N_ and *d*_N_ relative to a large diameter *D*_N_. The estimate of the <D>_v_ mean scattering region size (CSD) of NPs obtained from XRD patterns is shown in Appendix A. The values of <*D*>_v_ of both types of NPs agree with the average diameters <*D*_sphere_> within the calculation error of the latter (Appendix A), which confirms a high degree of crystallinity of the NPs. The CSD value is slightly higher than the average particle size calculated from microscopy data, probably due to the peculiarities of the size distribution (a significant fraction of large particles and bi-normal distribution). It is known [50] that a wide fraction of large particles increases the scatter of calculated CSD values, which can reach several nanometers in the case of particles larger than 15 nm.

The very close polydispersity values *Π_D_* ≈ *Π_d_ ≈* 0.7 (see Appendix B, Formula (A1)), calculated from large (*D*) and small (*d*) diameters, for both types of NPs indicate that the size distributions of NPs synthesized in different HTMW devices obey the same law. Large differences in the values of *Π* from unity indicate a wide size distribution of NPs in the colloidal system, which is a frequent phenomenon in the synthesis from an aqueous medium and is considered a disadvantage of the aqueous method of synthesis.

In addition to growing, NPs also aggregate and agglomerate with each other, which occurs as a result of their chaotic motion caused by temperature phenomena (Brownian motion, temperature gradients). Due to the processes of aggregation and agglomeration, both samples contain separate large NPs with a parameter *D* = 40–70 nm.

### 3.2. Excitation and Fluorescence Spectra

To select the excitation wavelength of luminescence when measuring the fluorescence kinetics, the fluorescence excitation spectrum of a 0.1 at% Nd^3+^: LaF_3_ type I NPs sample was measured at a detection wavelength of 862.8 nm by scanning the laser within the ^4^I_9/2_ → ^4^G_5/2_ + ^2^G_7/2_ electronic transition (Figure 4a) and detected at 1064 nm in the range of 300–950 nm (Figure 4b). There are two observed absorption maxima at 577.8 nm (Figure 4a) and 790 nm (Figure 4b, inset), respectively.

The luminescence of NPs samples of type I was excited at the ^4^I_9/2_ → ^4^G_5/2_ transition (λ_exc_ = 577.8 nm) (Figure 5a, orange arrow) and for NPs samples of type II at the ^4^I_9/2_ → ^4^F_5/2_ transition (λ_exc_ = 789 nm) (Figure 5a, red arrow). Note that for both types of NPs samples, the luminescence spectrum (Figure 6) and the luminescence decay kinetics from the ^4^F_3/2_ level do not depend on the above-mentioned excitation wavelengths. This is due to the fact that excitation, both from the ^4^G_5/2_ level, which is approximately 4500 cm^−1^ above the ^4^F_5/2_ level, and from the ^4^F_5/2_ level (Figure 5a), almost instantly (on a nanosecond time scale) nonradiatively relaxes (Figure 5b, blue arrows) to the metastable state ^4^F_3/2_.

The form-factors of the luminescence spectra of a 4 at% Nd^3+^: LaF_3_ type I NPs sample (Figure 6, red curve) and a 0.45 at% Nd^3+^: LaF_3_ single crystal (Figure 6, blue curve) are identical, which is in agreement with the HRTEM (Figure 1) and XRD (Appendix A) and indicates a good quality of crystallization of NPs in the obtained aqueous colloidal solutions. The positions of intense spectral lines, with maxima at about 860 and 862.8 nm for the ^4^F_3/2_(2’) → ^4^I_9/2_(1) and ^4^F_3/2_(1’) → ^4^I_9/2_(1) transitions, coincide in the excitation (Figure 6, black curve) and luminescence spectra of 0.1 and 4 at% Nd^3+^: LaF_3_ type I NPs samples. This result indicates that at room temperature the single type of optical centers in these NPs forms the Nd^3+^ ions spectral lines. At the same time, the spectral lines of the ^4^F_3/2_(2’) → ^4^I_9/2_(1) and ^4^F_3/2_(1’) → ^4^I_9/2_(1) transitions for NPs are better resolved, which indicates an even lesser inhomogeneous broadening of spectral lines when comparing to a single crystal.

### 3.3. Fluorescence Decay Kinetics, Relative Fluorescence Quantum Yield, and Brightness of Aqueous Colloidal Solutions

The fluorescence decay kinetics of the ^4^F_3/2_ metastable level of the Nd^3+^ ion in aqueous colloidal solutions of *x* at% Nd^3+^: LaF_3_ NPs was measured depending on the dopant concentration *x* = 0.1, 1, 2, 3, 4, 6, 8, 10, 12 within four orders of magnitude. Fluorescence detection was performed at the ^4^F_3/2_ → ^4^I_9/2_ transition at a wavelength λ_det_ = 862.8 nm upon laser excitation to the ^4^G_5/2_ + ^2^G_7/2_ level (λ_exc_ = 577.8 nm) for NPs samples of type I (Figure 7, blue curves) and to the ^4^F_5/2_ + ^2^H_9/2_ level (λ_exc_ = 789 nm) for NPs samples of type II (Figure 7, red curves). It was found that the measured luminescence kinetics of NPs samples of type II decays more slowly than the luminescence kinetics of NPs samples of type I at the same concentration *x* of Nd^3+^ ions.

To calculate the relative fluorescence quantum yield φ (Equation (4)), it is necessary to know the value of the spontaneous radiative lifetime $\tau_{R}$ of the ^4^F_3/2_ excited state of the Nd^3+^ ion. Theoretically, the spontaneous radiative decay time of the RE ion excited state in spherical NPs can be estimated using the approximate formula from paper [51]
(6)$$\frac{\tau_{R}^{nano}}{\tau_{R}^{bulk}}=\sqrt{\varepsilon }{\left(\frac{2+\varepsilon }{3}\right)}^{2}$$
where $\tau_{R}^{nano}$ and $\tau_{R}^{bulk}$ are the spontaneous radiative lifetime of the ^4^F_3/2_ level of Nd^3+^ ions in NPs and bulk crystal, respectively; $\varepsilon =\varepsilon_{cr}^{}/\varepsilon_{med}^{}=n_{cr}^{2}/n_{med}^{2}$ is the relative dielectric constant. The *n_cr_*(LaF_3_)= 1.593 [52] and *n_med_*(H_2_O) = 1.33 [53] are the refractive indices of the LaF_3_ bulk crystal and the medium (H_2_O) containing the crystalline NPs determined at a luminescence wavelength of 863 nm. Expression (6) is valid when the volume fraction of NPs in solution (in the medium) tends to zero *c* → 0. As follows from (6), in this limit, the value $\tau_{R}^{nano}$ of the ^4^F_3/2_ level of the Nd^3+^ ion in an aqueous colloidal solution of LaF_3_ NPs depends only on the value of $\tau_{R}^{bulk}$ in the LaF_3_ single crystal and the value of the parameter ε. If τRbulk(F3/24)=701 μs [17], then, in accordance with (6), the value of the spontaneous radiative lifetime in a spherical nanoparticle placed in an aqueous solution can be estimated as τRnano(F3/24)≈1100 μs.

Calculation of the spontaneous radiative lifetime for non-spherical NPs is a rather complicated theoretical problem that has not yet been solved. However, an analysis of some special cases carried out in [51] shows that the value of $\tau_{R}^{nano}$ may depend on the shape of the nanoparticle. Therefore, it should be expected that colloids with non-spherical NPs of the same crystal structure, but with different distribution functions with respect to deviation from sphericity, can have different radiation lifetimes.

The correct experimental determination of the $\tau_{R}$ value of the ^4^F_3/2_ level of the Nd^3+^ ion in aqueous colloidal solutions of Nd^3+^: LaF_3_ NPs is a separate problem. As it is known, at a sufficiently low level of excitation, there are two main decay channels for the excited electronic state of an RE ion introduced into the crystal matrix. One of them is due to intra-center processes: spontaneous emission and multiphonon relaxation. The other is associated with the transfer of the excitation energy to impurity optical centers, donors and acceptors of energy, which are randomly distributed in the system. If in the first channel (in the absence of inhomogeneous broadening of donor levels or latent anisotropy [54] associated with the non-sphericity of the NPs shape) quenching occurs exponentially with a characteristic time $\tau_{R}$, and in the second quenching channel the impurity quenching kinetics is substantially nonexponential.

Since the excitation relaxation channels are independent, the observed luminescence kinetics is the product of the kinetics of the excitation relaxation in each channel.
(7)$$I_{meas}(t)=N(t)\exp\left(-\frac{t}{\tau_{R}}\right)$$

The first determination method of $\tau_{R}$ logically follows from Formula (7) for $I_{meas}(t)$. According to it, it is necessary to synthesize a sample with the lowest possible concentration of impurities in such a way that the quenching channel could be neglected in the observed time interval: $N(t_{\max})\;\approx \;1$ ($\tau_{R}$ determination method 1). This method of $\tau_{R}$ determination was successfully applied, for example, for a 0.1 at% Nd^3+^: LaF_3_ single crystal [17], 0.05% Nd^3+^: YAlO_3_ [55], and phosphate glasses doped with Nd^3+^ [56].

However, when the system contains uncontrollable impurities, the energy acceptors, in addition to the activator, the situation becomes more complicated. In this case, no matter how we reduce the concentration of the activator, static quenching by uncontrolled acceptors remains. This situation is quite common in both nano- and bulk systems. In our case, these impurities are OH−acceptors randomly distributed in the volume of NPs. In a powder sample of Y^3+^ chelated complexes co-doped by Tb^3+^, the same OH−acceptors were contained in the structure of the crystal matrix [57], while in a bulk crystal activated by Nd^3+^ ions, this role was played by Dy^3+^ ions, which are effective quenchers of the ^4^F_3/2_ metastable state of Nd^3+^ ions [58]. Thus, if the rate of nonradiative energy transfer at the far stage of the fluorescence impurity quenching kinetics is comparable to the rate of radiative relaxation, the determination scheme should be changed. As shown in [34,56], OH−acceptors interact with excited Nd^3+^ ions via the dipole–dipole mechanism. If at the same time this distribution of acceptors in the volume of the nanoparticle is disordered, then all together should give the well-known Förster “square root” kinetics of luminescence quenching [59,60]. For further analysis of the kinetic data, we write down the theoretical formula in an explicit form
(8)$$I_{meas}(t)=\exp(-t/\tau_{R}-\gamma_{OH}\sqrt{t})$$
where
(9)$$\gamma_{OH}=\frac{4\pi^{3/2}}{3}n_{OH}\sqrt{C_{DA}^{OH}}$$

Here, $n_{OH}$ is the concentration of acceptors, and $C_{DA}^{OH}$ is the microparameter of the dipole–dipole donor–acceptor interaction.

If the quenching on acceptors is strong enough, then the second term in the exponent will be comparable to the first one or even dominate over the entire observation interval, leaving no room for the radiative decay exponent. In such a situation, we cannot determine $\tau_{R}$ according to the method 1. It is necessary to analyze a more general Formula (8). To determine $\tau_{R}$, it is necessary to build a function depending on $\sqrt{t}$. Then, for each selected value $\tau_{R}$ we analyze how the experimental data correspond to a straight line. With the optimal selection of the value $\tau_{R}$, we should obtain a straight line over the entire measurement interval, from the slope of which it is possible to determine the value of the static quenching macroparameter $\gamma_{OH}$ ($\tau_{R}$ determination method 2).
(10)$$\ln[N\left(t\right)]=\ln\left[I_{meas}(t)/\exp(-t/\tau_{R})\right]$$

For the practical implementation of the method 2, let us use logarithm kinetics (10) once again, presenting it in coordinates lg[−ln(*N*(*t*)] vs. lg*t*. In the general case, the slope angle of this dependence determines the power of time *t*, which in turn is determined by the multipolarity of the donor–acceptor interaction *s* and the dimension of the acceptor space *D* [61]. In our case, the interaction is dipole–dipole, i.e., *s* = 6, and the acceptors are located in the volume of NPs, which corresponds to *D* = 3 [19]. As a result, the power of *t* should be *s*/*D*= 3/6 = ½.

After proper selection of $\tau_{R}=1400\;\mu s$ for NPs of type I and $\tau_{R}=1300\;\mu s$ for NPs of type II (Figure 8), the impurity quenching kinetics for both types of NPs gives exactly a straight line with a slope of ½, which corresponds to the Förster static kinetics. As we can see, the values $\tau_{R}$ obtained using this procedure are 20% for type I and 15% for type II, higher than the theoretically estimated values made by Formula (6). This may be due to the non-spherical shape of the NPs. The difference in $\tau_{R}$ between the two studied samples is also not surprising. As we indicated above, this may be due to the difference in the distribution functions of NPs in colloids, with respect to deviation from sphericity.

Note that an attempt to determine $\tau_{R}$ directly from the slope of the luminescence kinetics at the far stage in the coordinates lg(*N*(*t*)) vs. *t* leads to significantly underestimated values of $\tau_{R}=805\;\mu s$ for the NPs sample of type I (Figure 8a) and $\tau_{R}=840\;\mu s$ for the NPs sample of type II (Figure 8b). As a result, after dividing $I_{meas}(t)$ by an exponential decay with such decay times $\tau_{R}$, we obtain not a monotonically decaying kinetics, but a curve growing at large times (blue curves in Figure 8a,b), which is unrelated to the disordered static stage of the impurity quenching kinetics.

Further, representing the fluorescence impurity quenching kinetics in special coordinates $\ln N(t)-\sqrt{t}$, we observe linearization of kinetics for both types of NPs in the entire time interval, except for the initial stage of ordered static decay. The slope of this stage in Figure 8 is close to unity. Having approximated the linear stage (Figure 9) for both types of samples, we obtained the following values for the macroparameters $\gamma_{OH}$ in 0.1 at.% Nd^3+^: LaF_3_ NPs: for type I $\gamma_{OH}^{I}=0.055\;{\mu s}^{-1/2}$ and for type II $\gamma_{OH}^{II}=0.041\;{\mu s}^{-1/2}$. Since the $\gamma_{OH}$ macroparameter linearly depends on the concentration of OH acceptors (Formula (9)), and $C_{DA}^{OH}$ is the same in both systems, it is possible to determine the ratio of their concentrations in types I and II samples as the ratio of the macroparameters themselves. It turns out that the number of OH−quenchers in the type II samples is 1.3 times less than in the type I samples.

In order to determine the absolute values of $n_{OH}$ and $C_{DA}^{OH}$, in addition to the $\gamma_{OH}$ value, it is also necessary to know the value of the boundary time $t_{1}$ between the exponential initial stage and the “root” Förster kinetics. By definition, the exponents of the ordered and disordered stages are equal at this point, whence the following relation follows for
(11)$$W_{ord}t_{1}=\gamma_{OH}\sqrt{t_{1}}\begin{array}{cc} ==> & t_{1}= \end{array}{\left(\gamma_{OH}/W_{ord}\right)}^{2}$$

In Formula (11), $W_{ord}$ is the rate of excitations quenching at the ordered stage.
(12)$$W_{ord}=n_{OH}\Omega C_{DA}^{OH}P$$

Here, Ω is the volume per one site of the acceptor sublattice, and P is the lattice sum in the case of dipole–dipole interaction of the donor with acceptors uniformly distributed over the corresponding sublattice.
(13)$$P=\sum_{k}{\left(1/r_{k}\right)}^{6}$$

Substituting expressions (9) and (12) into (11), we obtain the following expression for the boundary time:(14)$$t_{1}={\left(\gamma_{OH}/W_{ord}\right)}^{2}=\frac{16\pi^{3}}{9\Omega^{2}P^{2}C_{DA}^{OH}}$$

It is easily seen from this relationship that $t_{1}$ depends only on the structural constants of the acceptor sublattice and the microparameter of the dipole–dipole donor–acceptor interaction $C_{DA}^{OH}$. Therefore, having determined the value of $t_{1}$, we can immediately calculate the value of $C_{DA}^{OH}$ using the formula
(15)$$C_{DA}^{OH}=\frac{16\pi^{3}}{9\Omega^{2}P^{2}t_{1}}$$

A boundary time $t_{1}$ is equal to the abscissa of the asymptotes intersection of the corresponding stages, with slope angles of 1 and ½ (Figure 8). Accordingly, for both systems type I and II, we obtain $t_{1}^{}\approx 15\;\mu s$. It is assumed that the hydroxyl OH^−^ group replaces the F^−^ ion in the crystal lattice of LaF_3_. The corresponding lattice sum and the volume per fluorine site are equal [62]
(16)$$P_{La-F}=\text{44,793}\;{\mathrm{nm}}^{-6}\begin{array}{cc} , & \Omega_{F}= \end{array}1/n_{F\max}^{}\approx 1/55.05\;{\mathrm{nm}}^{3}$$

Further, using the value $t_{1}$ and calculating $C_{DA}^{OH}$ according to Formula (15), we obtain
(17)$$C_{DA}^{OH}\approx 0.0056{\mathrm{nm}}^{6}/\mathrm{ms}$$

Now, knowing $C_{DA}^{OH}$, $\gamma_{OH}$ and using (9), we can determine the absolute value of the acceptors concentration $n_{OH}^{I}\approx 3.15\;{\mathrm{nm}}^{-3}$, $n_{OH}^{II}\approx 2.33\;{\mathrm{nm}}^{-3}$. Then, for a dimensionless relative concentration, $c_{OH}=n_{OH}\Omega_{F}\times 100\%$, we obtain $c_{OH}^{I}=5.7\%$ and $c_{OH}^{II}=4.2\%$, respectively, which are rather large values. It shows that diffusion of OH^−^ ions in the lattice with the subsequent exchange for $F^{-}$ ions during hydrothermal crystallization of precipitated gels is difficult. Most likely, an increase in the duration and temperature of hydrothermal treatment should lead to a decrease in the concentration of defects in resulting NPs.

By integrating the luminescence decay kinetics and dividing it by $\tau_{R}=1400\;\mu s$ for the NPs of type I series and by $\tau_{R}=1300\;\mu s$ for the NPs of type II series, we can calculate the fluorescence relative quantum yield (Formula (4)). Figure 10a shows the concentration dependence of the fluorescence relative quantum yield for both series of NPs samples in comparison with a concentration series of single crystals. The difference in the fluorescence relative quantum yield of aqueous colloids of NPs is that it does not tend to be 100% in the limit of low values of the Nd^3+^ concentration, as it was in bulk crystals. The characteristic cutoff of the concentration dependence at *ϕ* = 26–33% indicates a high contribution of luminescence quenching on OH−acceptors.

The maximum relative quantum yield of luminescence of aqueous colloidal solutions, associated with quenching on third-party acceptors only, as expected, is observed at the minimum concentration of Nd^3+^ ions (0.1 at.%) and for type II samples exceeds 30%. This value is even larger than for dried powder obtained by the HTMW treatment with the surfactant Proxanol-268 (Proxanol-268 surfactant) [34] and is undoubtedly a record result for stable aqueous colloids of NPs doped with Nd^3+^ ions. The concentration dependences of the fluorescence relative quantum yield φ (Equation (4)) and the fluorescence brightness *ν* (Equation (5)) of the two types of NPs samples differ from each other over the entire range of Nd^3+^ concentrations. Due to systematic and random errors in the preparation of initial salt solutions during the synthesis of aqueous colloidal solutions of NPs, the concentration of impurity Nd^3+^ ions in them may slightly vary. Therefore, for better verification of the φ and *ν* values, we performed repeated syntheses of both samples of NPs solutions with the same Nd^3+^ concentration by inclusion. The dispersion in the φ and *ν* values for both series of NPs samples with nominally the same Nd^3+^ concentration is indicated by double and triple circles of the same color (Figure 10).

Our analysis showed that the lower value of φ in the sample of NPs of type I compared to NPs of type II is explained by the higher concentration of uncontrolled impurity of OH−groups in NPs of type I. Effective quenching of the excitation of the ^4^F_3/2_ level of Nd^3+^ ions in Nd^3+^: LaF_3_ NPs is caused by the interaction with anharmonic vibrations of OH−groups [34]. During the hydrothermal synthesis, the hydroxyl OH−group randomly replaces F^–^ ions in the entire volume of the nanoparticle, which is responsible for the Förster kinetics of luminescence quenching at low neodymium concentrations. With an increase in the concentration of Nd^3+^ ions in NPs, self-quenching appears, where the role of acceptors is played by unexcited Nd^3+^ ions, and at the same time, the resonant migration of excitations over impurity sites of the activator significantly accelerates both self-quenching and quenching of excitations on additional acceptors.

When migration-accelerated self-quenching prevails over radiative decay (strong self-quenching: $V_{DA}n_{D};V_{DD}n_{D}>>1$) in the case of a dipole–dipole donor–acceptor interaction, the theory [17,63] gives the following expression for the brightness:(18)$$\nu =\frac{n_{D}}{1+V_{DA}\left(V_{DA}+V_{DD}\right)n_{D}^{2}}$$

Here, $n_{D}$ is the donor concentration, and the effective volumes of strong incoherent donor–acceptor are $V_{DA}$ and donor–donor, $V_{DD}$, interactions, which are determined through the corresponding Förster radii:(19)$$V_{DA}=\frac{\pi^{}}{2}\frac{4\pi }{3}\left(R_{DA}^{3}=\sqrt{C_{DA}\tau_{R}}\right),\begin{array}{cc} & V_{DD} \end{array}=\frac{\pi^{}}{2\sqrt{2}}\frac{4\pi }{3}\left(R_{DD}^{3}=\sqrt{C_{DD}\tau_{R}}\right)$$

Förster radii of the strong interaction are determined in the usual way from the condition of the equality of the rates of radiative and nonradiative decay of excitation at the critical distance
(20)$$w_{DA}(R_{DA}^{})=\frac{C_{DA}}{R_{DA}^{6}}=\frac{1}{\tau_{R}},\begin{array}{cc} & w_{DD}(R_{DD}^{})=\frac{C_{DD}}{R_{DD}^{6}}=\frac{1}{\tau_{R}} \end{array}$$

As a rule, in systems doped with rare earth ions, a hopping mechanism is realized in which the inequality $R_{DD}^{}>>R_{DA}^{}$.

Formula (18) gives the concentration dependence of the luminescence brightness during self-quenching, without any quenching on additional acceptors. However, if quenching occurs only on independent acceptors randomly distributed in a system with the concentration $n_{A}$ and no self-quenching process takes place, then the theory of migration-accelerated quenching [64,65] gives a completely different result. In the same simple form that is valid for strong hopping quenching, the brightness for dipole–dipole interactions is equal to
(21)$$\nu =\frac{n_{D}}{1+2V_{DA}^{OH}n_{A}\left(V_{DA}^{OH}n_{A}+V_{DD}^{}n_{D}^{}\right)/\pi },$$

It is easy to see from Formula (18) that in the case of pure self-quenching, the concentration dependence of the brightness initially has a linear increase, proportional to $n_{D}$, and then, passing through a maximum, at high concentrations decreases proportionally to $1/n_{D}$. In the case of pure quenching (21), the brightness at high concentrations reaches a plateau of $\nu \sim \pi /(2V_{DA}^{OH}V_{DD}^{}n_{A})$.

In our experiments, the maximum fluorescence brightness ν in the series of single crystals is observed at a Nd^3+^ ions concentration of about 1.3 at% and for that of NPs of type II at about 1.8 at%, while in the series of NPs of type I, at the concentrations of more than 1.8 at%, the brightness practically does not change (Figure 10b). At the same concentrations of Nd^3+^ ions of the NPs sample of type II, the fluorescence relative quantum yield and fluorescence brightness are only 2.5 times lower than in similar Nd^3+^: LaF_3_ single crystals and 20% higher than in samples of NPs of type I.

Thus, in agreement with Formula (18) in a bulk crystal, pure self-quenching with a pronounced maximum is observed (Figure 10b, black dots). For the series of NPs of type I, on the contrary, a plateau is observed (Figure 10b, blue dots), which indicates the prevalence of the quenching over the self-quenching, in agreement with Formula (21). Finally, the concentration dependence of the brightness of the series of NPs of type II has a weakly pronounced maximum (Figure 10b, red dots), shifted relative to the single crystal towards higher concentrations. This means that an intermediate situation of two-channel quenching is realized in this system, when both self-quenching and quenching make comparable contributions.

By the combination of the properties of our NPs: (1) “green” water synthesis, (2) short synthesis duration, (3) long-term stability of an aqueous colloid within several months without observable sedimentation of NPs [19], (4) no additional modification of the NPs surface, (5) high luminescence brightness in the first biological transparency window, and (6) low cytotoxicity [21,66], the aqueous colloidal solutions of Nd^3+^: LaF_3_ NPs of type II surpass the similar aqueous colloidal solutions of Nd^3+^: LaF_3_ NPs described in the literature.

## 4. Conclusions

In this work, two concentration series of long-term stable aqueous colloidal solutions of Nd^3+^: LaF_3_ crystalline NPs that possess a fluorcerite structure were synthesized by the HTMW method in the experimental setups with either one or two magnetrons. The Nd^3+^: LaF_3_ NPs resulting from these two synthetic setups are single-phase, well crystallized, morphologically different, and partially faceted with an average size of 15.4 ± 6 nm (NPs of type I) and 21 ± 7 nm (NPs of type II). Both types of NPs have a size distribution that can be described by a double Gaussian function. The finely dispersed fraction of NPs is apparently formed as a result of the primary crystallization of the gel from an aqueous solution. The fraction of larger NPs is probably the result of recrystallization and the growth of primary NPs during the HTMW treatment. In the setup with two magnetrons, it seems that a more uniform supply of microwave radiation to the autoclave containing the mother liquor is realized. This increases the rate of growth and recrystallization of type II NPs, leading to an increase in their average size and degree of crystallinity.

Differences in morphologies and in size distributions of NPs affect their physical properties. The non-spherical shape of NPs, different size distribution functions, and different concentration of OH−groups that randomly replace F^–^ ions in the entire NP volume all primarily affect the physical parameters that determine the relative quantum yield of fluorescence and their relative fluorescence brightness. Namely, the radiative lifetime $\tau_{R}$ and macroparameters of luminescence quenching $\gamma_{OH}$, which turned out to be different in NPs of different types.

This paper describes in detail the methodology for determining the radiative lifetime and macroparameter of luminescence quenching in NPs, which is more complicated in comparison with the corresponding bulk crystals. The main reason for this complication is high concentrations of OH−acceptors, which explains why, even at low concentrations of Nd^3+^ ions (0.1 at%) when self-quenching is absent and the luminescence kinetics can be represented in the form (8), the fluorescence quenching on additional acceptors dominates over radiative decay in the observed time interval. To determine $\tau_{R}$ in such situations, it is necessary to plot the function $N(t)=I_{meas}(t)/\exp(-t/\tau_{R})$ in special coordinates and to select the values of $\tau_{R}$ in such a way that the function N(t) over the entire observation time interval fits a straight line, corresponding to the Förster kinetics with a slope angle of ½ (Figure 8). Having checked the functional dependence in time in this way, it is then necessary to rearrange the kinetics in the coordinates of Figure 9, in which the kinetics should have the form of a straight line with a tangent slope equal to the macroparameter $\gamma_{OH}$. With an optimal choice of the $\tau_{R}$ value, it is possible to obtain a straight line over the entire measurement interval and then to determine the static quenching macroparameter $\gamma_{OH}$ from the angle of its slope.

With an increase in the concentration of Nd^3+^ ions, the migration of excitations and the self-quenching appear in the system of these impurity centers. Migration accelerates both channels of excitation relaxation competing in NPs, self-quenching (Nd* → Nd), and quenching (Nd* → OH). Self-quenching itself, which occurs in single crystals, gives a distinct maximum in the dependence of the luminescence brightness on concentration, as described by Equation (18) (Figure 10b, black dots). On the contrary, the simple quenching on additional acceptors leads to the exit of the luminescence brightness curve to a plateau (Equation (21)). In NPs, these two channels compete with each other. Therefore, in NPs of type I, where quenching prevails over self-quenching, a plateau is observed (Figure 10b, blue dots). At the same time, in NPs of type II where the concentration of OH^–^ acceptors is lower, both excitation relaxation processes have a comparable contribution, and the brightness maximum smooths and shifts towards higher concentrations of Nd^3+^ ions (Figure 10b, red dots). A decrease in the relative quantum yield and fluorescence brightness of the ^4^F_3/2_ level of Nd^3+^ ions in aqueous colloidal solutions of Nd^3+^: LaF_3_ NPs in comparison with similar single crystals containing the same concentrations of impurity Nd^3+^ ions is caused exactly by the interaction with anharmonic vibrations of OH-groups. Therefore, the conclusion of the work on the effect of the number of magnetrons on the concentration of OH-groups in the volume of NPs is important, since it provides a basis for optimizing the luminescent properties of aqueous colloidal solutions of NPs. The relative quantum yield of fluorescence and fluorescence brightness of an aqueous colloidal solution of the NPs of type II are only 2.5 times lower than that of the analogous Nd^3+^: LaF_3_ single crystals. This property offers promising prospects for the use of these colloidal solutions for bioimaging.

The results obtained on the fluorescence of our NPs show that aqueous colloidal solutions of 2 at% Nd^3+^: LaF_3_ NPs of type II synthesized on a setup with two magnetrons are more promising for biological imaging, since their fluorescence brightness is about 25–30% higher than solutions of NPs of type I synthesized in a setup with a single magnetron.

## Figures and Tables

**Figure 1 nanomaterials-11-02847-f001:**
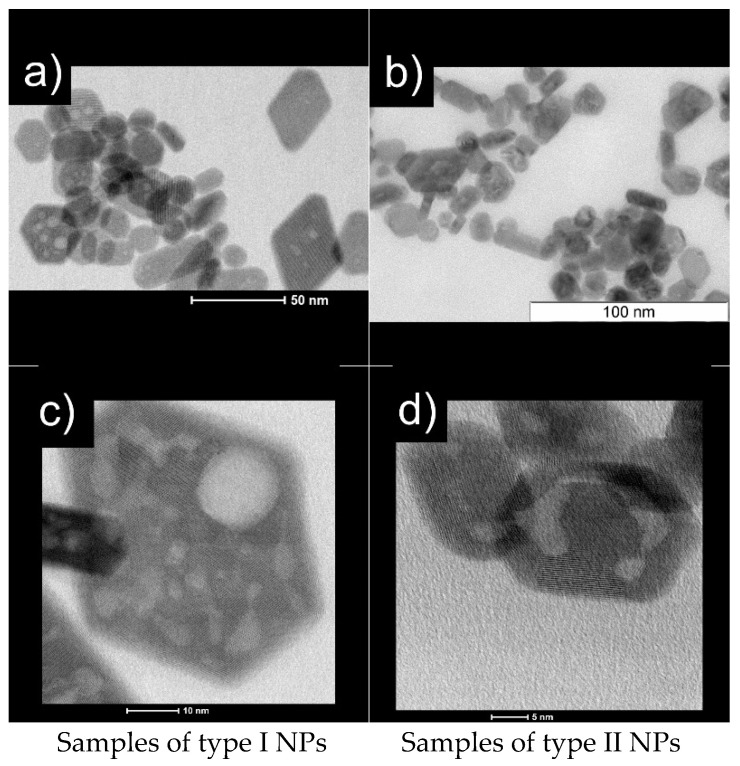
STEM images of Nd^3+^: LaF_3_ NPs: samples of type I are on the left (**a**,**c**); samples of type II are on the right (**b**,**d**).

**Figure 2 nanomaterials-11-02847-f002:**
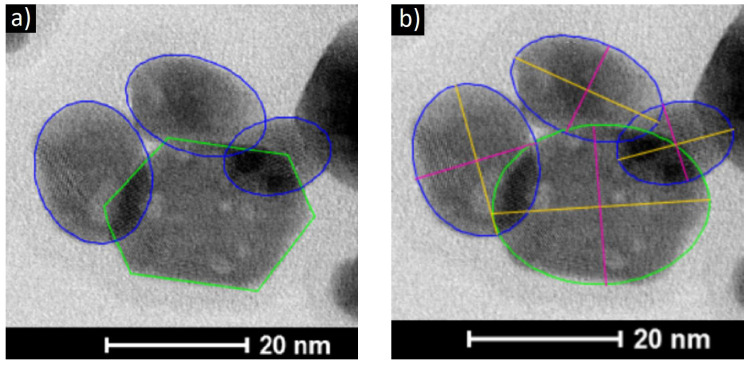
Outlined and approximate in shape individual nanoparticles (blue contours-rounded non-faceted NPs (fine fraction), greenish elongated faceted hexagons of NPs (coarse fraction))—(**a**); linear dimensions of NPs (yellow axis-large diameter (D) of the ellipse, pink axis-small diameter (d) of the ellipse)—(**b**).

**Figure 3 nanomaterials-11-02847-f003:**
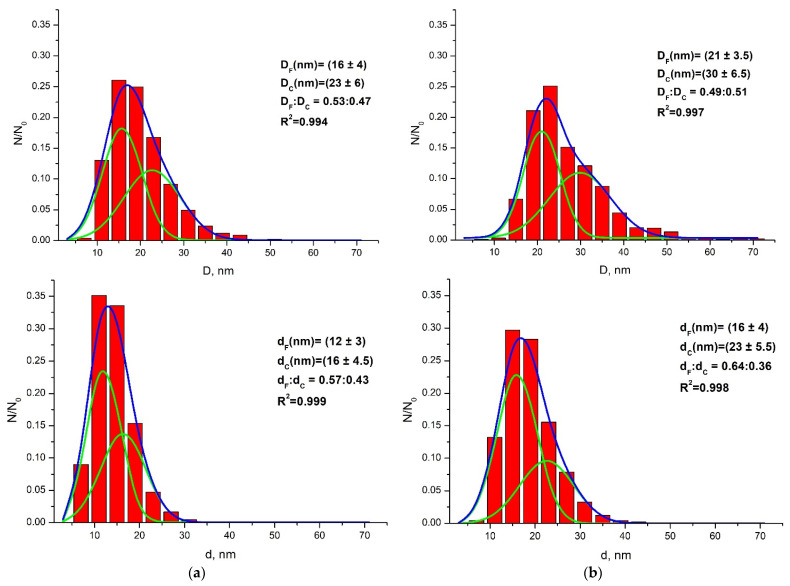
General distribution of NPs of samples of type I (**a**) and type II (**b**) by characteristic sizes, in accordance with fine (Fine, index F) and coarse (Coarse, index C) fractions; the shape of the nanoparticle projections (in TEM images) is approximated by model ellipses with large (*D*) and small (*d*) diameters; parameters *D*_F_, *d*_F_ and *D*_C_, *d*_C_ are the maxima of green curves described by Gaussian functions, the sum of which (blue curve) describes the size distributions of these two fractions of small and large Nd^3+^: LaF_3_ NPs.

**Figure 4 nanomaterials-11-02847-f004:**
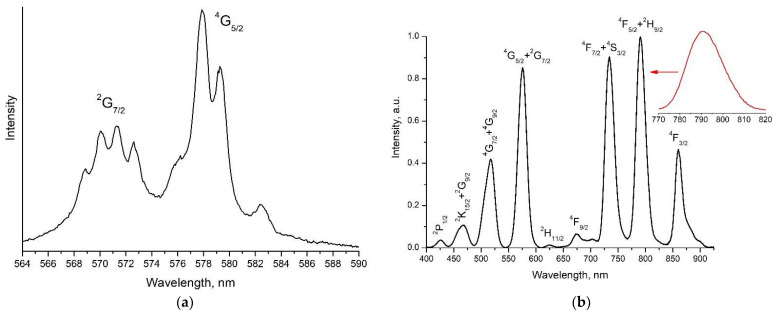
Fluorescence excitation spectrum of the 0.1 at% Nd^3+^: LaF_3_ type I NPs sample detected at 862.8 nm when scanning the laser within the ^4^I_9/2_ → ^4^G_5/2_ + ^2^G_7/2_ transition wavelengths, normalized to the laser energy (**a**) and detected at 1064 nm in the range of 400–925 nm (**b**).

**Figure 5 nanomaterials-11-02847-f005:**
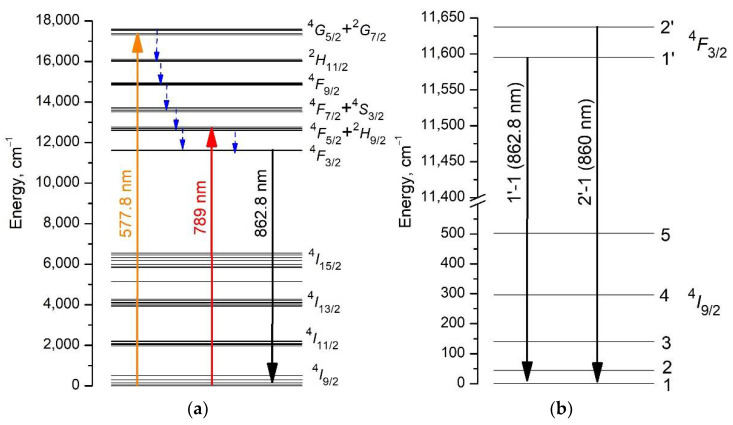
Energy level diagram of the Nd^3+^ doped LaF_3_ crystal (on the left) [35]: luminescence excitation transitions ^4^I_9/2_ → ^4^G_5/2_ + ^2^G_7/2_ (orange arrow) and ^4^I_9/2_ → ^4^F_5/2_ + ^2^H_9/2_ (red arrow) and fluorescence transition ^4^F_3/2_ → ^4^I_9/2_ (black arrows). Blue dashed arrows indicate multiphonon relaxation of electronic excitations. (**a**) Crystal field splitting levels (Stark levels) of the ^4^I_9/2_ and ^4^F_3/2_ manifolds of Nd^3+^ ion in the Nd^3+^: LaF_3_ crystal and (**b**) optical inter-manifold transitions between Stark levels 1’-1 and 2’-1 are shown.

**Figure 6 nanomaterials-11-02847-f006:**
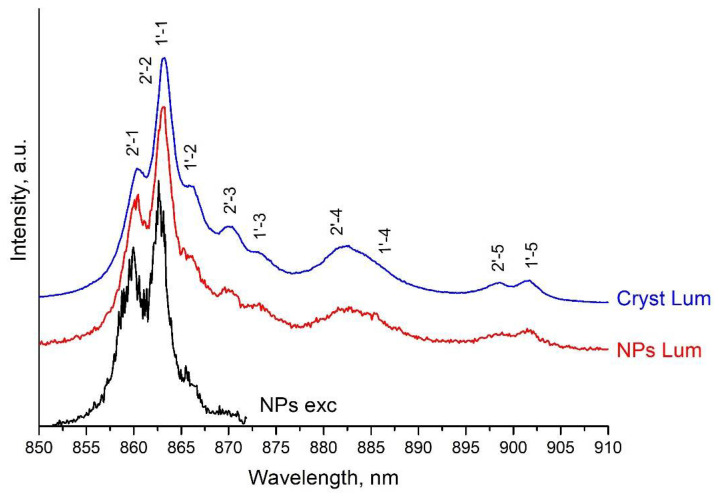
The luminescence excitation spectrum of a 0.1 at% Nd^3+^: LaF_3_ NPs sample of type I (black spectrum), measured by scanning a laser at the ^4^I_9/2_ → ^4^F_3/2_ transition with a step of 0.12 nm. The detection was performed in the range of 850–873 nm, including ^4^F_3/2_ (j’) → ^4^I_9/2_ (j) transitions, where j = 1 and 2. The luminescence spectra of a 4 at% Nd^3+^: LaF_3_ NPs sample of type I (red spectrum) and 0.45 at% Nd^3+^: LaF_3_ single crystal (blue spectrum) upon excitation at a wavelength λ_exc_ = 577.8 nm, measured at the ^4^F_3/2_ → ^4^I_9/2_ transition.

**Figure 7 nanomaterials-11-02847-f007:**
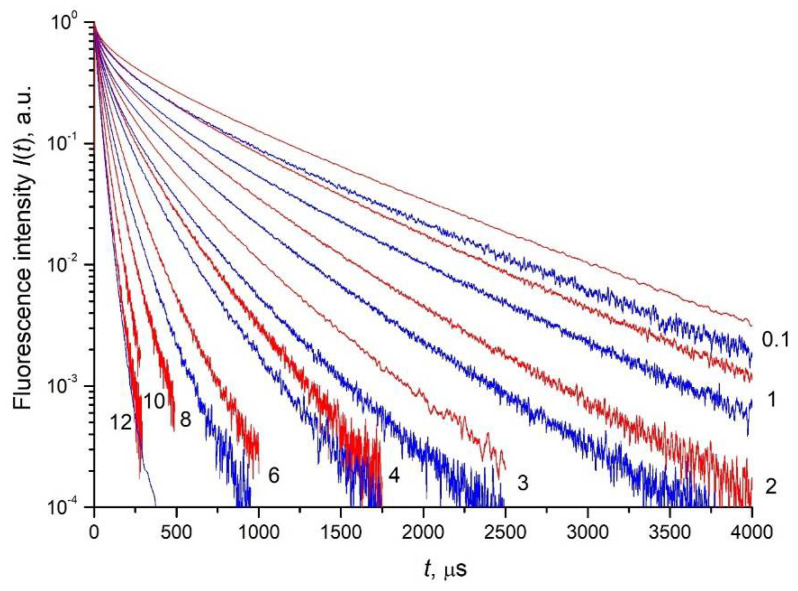
Luminescence decay kinetics of *x* at% Nd^3+^: LaF_3_ (*x* is indicated in the figure) detected at the ^4^F_3/2_ → ^4^I_9/2_ transition of the Nd^3+^ ion at λ_det_ = 862.8 nm wavelength upon laser excitation to the ^4^G_5/2_ level (λ_exc_ = 577.8 nm) for NPs samples of type I (blue curves) and to the ^4^F_5/2_ + ^2^H_9/2_ level (λ_exc_ = 789 nm) for NPs samples of type II (red curves).

**Figure 8 nanomaterials-11-02847-f008:**
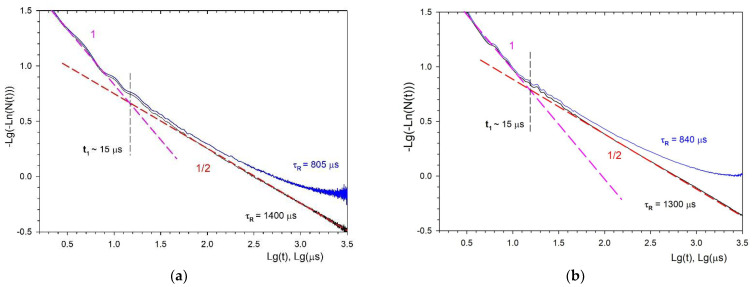
Fluorescence impurity quenching kinetics $N(t)=I_{meas}(t)/\exp(-t/\tau_{R})$ (black curves) from the ^4^F_3/2_ level of the Nd^3+^ ion in NPs of type I (**a**) and type II (**b**) of an aqueous colloidal solution of 0.1 at.% Nd^3+^: LaF_3_ NPs at $\tau_{R}=1400\;\mu s$ for NPs of type I and at $\tau_{R}=1300\;\mu s$ for NPs of type II. The blue curve is N(t) when trying to approximate the late stage of luminescence decay kinetics $I_{meas}(t)$ by an exponential function $\exp\left(-t/\tau_{R}\right)$ at (**a**) $\tau_{R}=805\;\mu s$ for NPs of type I and (**b**) at $\tau_{R}=840\;\mu s$ for NPs of type II. Red dashed lines are Förster kinetics $\exp\left(-0.055\sqrt{t}\right)$ and $\exp\left(-0.041\sqrt{t}\right)$. The magenta dashed lines are the ordered decay stage $\exp\left(-W_{ord}t\right)$.

**Figure 9 nanomaterials-11-02847-f009:**
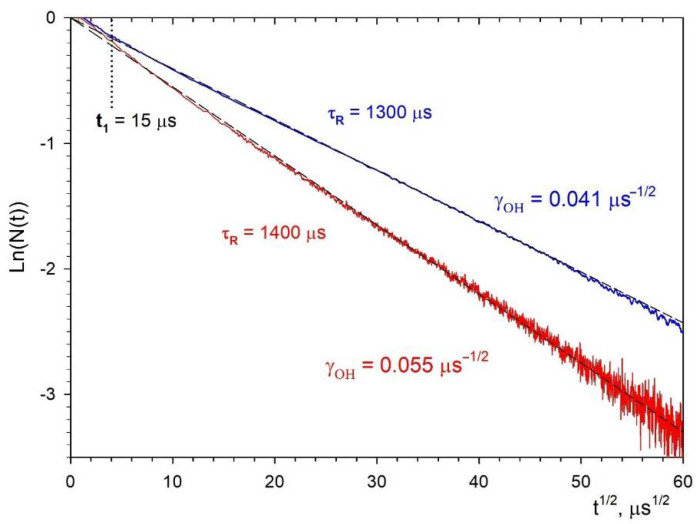
The static quenching kinetics of luminescence of the ^4^F_3/2_ level of the Nd^3+^ ion for an aqueous colloidal solution of 0.1% at.% Nd^3+^: LaF_3_ NPs: type I (red curve) and type II (blue curve). Black dashed lines are Förster kinetics $N(t)=\exp\left(-0.055\sqrt{t}\right)$ and $N(t)=\exp\left(-0.041\sqrt{t}\right)$.

**Figure 10 nanomaterials-11-02847-f010:**
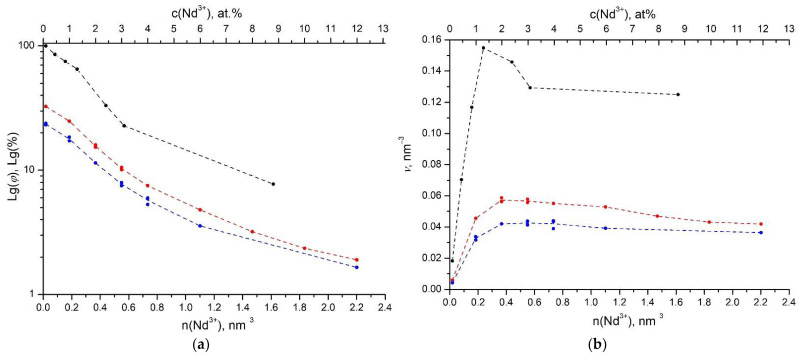
Concentration dependence of the relative quantum yield φ (**a**) and fluorescence brightness *ν* (**b**) of the ^4^F_3/2_ level of the Nd^3+^ ion in single crystals (black circles) and aqueous colloidal solutions of Nd^3+^: LaF_3_ NPs samples of type I (blue circles) and type II (red circles) (for NPs samples of type I $\tau_{R}=1400\;\mu s$ and NPs of type II $\tau_{R}=1300\;\mu s$ ). Dashed blue and red curves are drawn through the arithmetic mean values of the corresponding parameters. For single crystals, the dashed black curve is drawn through the measured values of φ and *ν*.

## Data Availability

The data presented in this study are available on request from the corresponding author.

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
