# Peer review of "Stable Aqueous Colloidal Solutions of Nd3+: LaF3 Nanoparticles, Promising for Luminescent Bioimaging in the Near-Infrared Spectral Range"

_nanomaterials, 2021, doi:10.3390/nano11112847_

Round 1
Reviewer 1 Report
In the manuscript by Popov et al. Nd3+:LaF3 nanoparticles are synthesized in different manners to evaluate the effect of Nd-Nd and -OH oscillations on the luminescent quenching aiming at understanding the way to increase the NIR luminescence performance for bioimaging applications. Dwelling the Judd-Ofelt calculations in details for Nd-containing objects characterize this study as a good theoretical work supported by reasonable number of experimental executions. In general, this manuscript can be of interest to academics working in the field of lanthanide chemistry and nanomaterials. However, several issues should be carefully addressed before publishing in Nanomaterials.
- What is the specific meaning of AQS acronym? Should it be ACS?
- The paper should be subjected stringent spell-check. Some of the typos: Ptovides, value4 etc.
- At page 4 it is claimed that “the almost complete absence of the luminescence quenching of the Nd3+ ions by OH acceptors in the bulk of NPs obtained by a non-aqueous method”. What about the surface Nd3+ ions? Is it correct to compare 30 nm aqueous-ethanol NPs with 10 nm non-aqueous polymer coated ones? The best way be to perform the same syntheses in deuterated solvents.
- The introduction section is too cumbersome. It is recommended to amend well-known processes. For instance, “During the initial dissolution of the gel, two competing processes occur, depending on the local supersaturation of the solution – the formation of nuclei and the growth of already formed nuclei. At the first stage, the nucleation process predominates and many nuclei are formed on the surface of the dissolving gel. Then the formed centers begin to grow due to the dissolving gel. In addition to their growth in the mother medium, there are also particle aggregation processes that occur as a result of chaotic movement of NPs caused by Brownian motion and motion associated with temperature gradient fields. With the accumulation of primary crystallites, the next stage of the process begins, at which the growth and recrystallization of the formed particles occurs (dissolution of small and highly defective particles and the growth of large well-crystallized particles), which determines the concentration of defects and the degree of crystallinity of NPs [42,43].” text can easily be substituted by “collective recrystallization” term. 4.5 pages are exceedingly much. It is recommended to squeeze it focusing on goal setting and novelty/originality emphasis.
- Topic related and field important fresh articles (10.1016/j.msec.2019.110057 and 10.1007/s10853-019-03532-6) should be referenced to strengthen the beginning of introduction section.
- It is recommended to unify scalebars at Figure 1. Scalebars on Figures 1c and 1d are poorly readable, please enlarge. Four images should be collected in a neat four-panel illustration. Negligently scattered images make this text non-friendly to reader.
- Talking about colloidal stability would be of great importance to refer DLS data and zeta-potential in solutions.
- For better understanding all bands at Figure 4b should be assigned with corresponding transitions (likewise Figure 4a).

Author Response
Thank you very much for the appreciation of our work and helpful comments. Note that this work is devoted to the development of the methods for determining the exact values of the relative quantum yields of fluorescence and its brightness, which requires a correct determination of the radiative lifetime of NPs doped with rare earth ions in the presence of quenching of OH acceptors in analysis of fluorescence kinetics decay. Comprehensive analysis using Judd-Ofelt theory of the fluorescence emission rates and fluorescence quenching in rare earth ions doped NPs were done in our previous paper [Alloys and Compounds 2018, 756, 182–192. doi.org/10.1016/j.jallcom.2018.05.027]
- We have changed AQS acronym to ACS.
- We made the spell checking.
- The remark (3) is not very clear for us. In the manuscript, we are trying to explain that the fluorescence quenching for the NPs in the nanocomposite is stronger than it is claimed by the authors of paper [30]. We are not trying to compare the NPs synthesized by different methods and completely agree with the reviewer that it is incorrect to compare 30 nm aqueous-ethanol NPs with 10 nm non-aqueous polymer coated. Indeed, for small NPs of 10 nm in diameter and high concentration of dopant ions the fluorescence quenching caused by vibrations of molecular groups located on the surface should be accounted. However, here in the introduction we just focus on the correct determination of the radiative lifetime. We have changed introductory words “At the same time, “… to “Similary,….” to avoid misunderstanding.
- We have changed the text in the Introduction according to reviewer’ suggestions.
When a freshly precipitated gel is exposed to hydrothermal conditions, it undergoes a collective recrystallization process, which largely determines the size distribution of nanoparticles, the stability of final colloid, and the degree of crystallinity and the defectiveness of NPs. [42,43 44,45]. [42,43 44,45]. During the initial dissolution of the gel, two competing processes occur, depending on the local supersaturation of the solution – the formation of nuclei and the growth of already formed nuclei. At the first stage, the nucleation process predominates and many nuclei are formed on the surface of the dissolving gel. Then the formed centers begin to grow due to the dissolving gel. In addition to their growth in the mother medium, there are also particle aggregation processes that occur as a result of chaotic movement of NPs caused by Brownian motion and motion associated with temperature gradient fields. With the accumulation of primary crystallites, the next stage of the process begins, at which the growth and recrystallization of the formed particles occurs (dissolution of small and highly defective particles and the growth of large well-crystallized particles), which determines the concentration of defects and the degree of crystallinity of NPs [42,43].
- Thank you for not previously seen interesting works about non-toxic NPs emitting in the NIR spectral range. We referenced them in the introductory section.
- At your request, we have changed Figure 1 to an image montage. It is not possible to put the same scale marks, since the micrographs were taken with different microscopes and different It is not entirely correct to make changes to the original data.
- It was not a goal of this manuscript to analyze the reason of very good stability of our aqueous colloidal solutions. This will be done in our next study, where we are going to consider also the individual fluorescence properties of single nanoparticles. In this manuscript, this would complicate the perception of the main idea of the article, associated with the correct method for determining the relative quantum yield of luminescence and its brightness, and analyzing its impurity concentration dependence.
- We have improved Fig. 4b according to reviewer’s recommendation.

Reviewer 2 Report
The work can be accepted because the study is complete, but the conclusions can be summarized much more.

Author Response
Thank you for the high appraisal of our work. We regard the comment of Reviewer 2 as a non-binding recommendation for changes. We think that such a detailed conclusion gives the reader a better understanding of the main theoretical and experimental results of the study and emphasizes their agreement.

Reviewer 3 Report
Paper sounds well, especially for the discussion aboyt the relaxation mechanism. I have appreciated the results, discussed with competence, as well as the general presentation of the paper

Author Response
Many thanks for the appreciation of our work.

Round 2
Reviewer 1 Report
With careful addressing of the comments and revisions brought in the text this manuscript is recommended now to be accepted in the present form.